# The Impact of Short-Term Shark Liver Oil Supplementation on the Fatty Acid Composition of Erythrocyte Membranes

**DOI:** 10.3390/nu13103329

**Published:** 2021-09-23

**Authors:** Katarzyna Zakrzewska, Katarzyna Oszajca, Wojciech Zep, Anna Piekarska, Malgorzata Sidorkiewicz

**Affiliations:** 1Department of Medical Biochemistry, Medical University of Lodz, Mazowiecka 6/8, 92-215 Lodz, Poland; katarzynaz90@onet.eu; 2Medical Committee of Polish Football Association, 02-366 Warsaw, Poland; wojciechzep@gmail.com; 3Department of Infectious Diseases and Hepatology, Medical University of Lodz, 91-347 Lodz, Poland; anna.piekarska@umed.lodz.pl

**Keywords:** erythrocyte fatty acids, shark liver oil, PUFAs, omega-3, DHA, C-reactive protein, cholesterol, LDL-C, HDL-C

## Abstract

Fatty acid (FA) balance is strictly related to human health. The composition of fatty acids in lipid membranes seems to be influenced by diet. Shark liver oil (SLO) supplementation has been widely used recently in the prevention and treatment of human diseases. We analyzed the impact of short-term SLO supplementation on certain biochemical parameters and erythrocyte FA composition in a group of young healthy women. Our results showed that 6 weeks of SLO supplementation led to a significant decrease in C-reactive protein levels in sera and intracellular cholesterol levels in peripheral blood mononuclear cells. SLO supplementation caused a significant increase in the content of the polyunsaturated omega-3 FAs: docosahexaenoic acid, docosapentaenoic acid and α-linolenic acid. In the group of omega-6 FAs, we observed a significant elevation of arachidonic and dihomo-gamma-linoleic acid content. Due to these alterations, the omega-3 index increased significantly from 3.6% (before) to 4.2% (after supplementation). We also observed the impact of SLO supplementation on the membrane fluidity index. The ratio between saturated and unsaturated FAs decreased significantly from 13.1 to 9.9. In conclusion, our results show that even short-term SLO supplementation can improve human erythrocyte fatty acid composition and other parameters that may have health-promoting consequences.

## 1. Introduction

Fatty acid (FA) balance is intrinsically involved in maintaining metabolic processes important for normal body function. FAs serve as energy sources, signaling molecules and irreplaceable essential components of cellular membranes. Following the results of recent studies, we should consider FAs first and foremost to be the basic unit of biological membranes and, secondarily, a source of energy [1]. An increasing body of experimental and epidemiological evidence suggests that the proper ratio between saturated and polyunsaturated fatty acids (PUFAs) plays a relevant role in the maintenance of the functionality of cells [2,3]. PUFAs are functional and structural components of cell membranes and in the structure of phospholipids, they are present as acyl chains primarily esterified at the sn-2 position of glycerol. Depending on the position of the terminal double bond, they are grouped into omega-3 and omega-6 FAs. Different effects of omega-3 and omega-6 on membrane properties indicate the importance of the balance between these two FAs in determining the function of biological membranes. Any perturbation of FAs homeostasis may modulate membrane function. The measurement of erythrocyte FAs levels may indirectly reflect the FAs status in the whole body [4]. The composition of erythrocyte FAs is an important factor that modulates the membrane fluidity, which is measured by the ratio of saturated to unsaturated FAs. This means that FAs have implications for membrane fluidity and therefore also cellular function [5]. In addition, the omega-3 index, defined as a sum of the relative percentages of eicosapentaenoic (EPA) and docosahexaenoic (DHA) acids to the total FAs in erythrocyte membranes is the frequently used measure to reflect the intake of the long-chain omega-3 FAs. An index value of 2–3% indicates low EPA and DHA intake, whereas 11–12% indicates a high intake [6,7]. An omega-3 index of 8% was described as having a low risk of cardiac events. This level is achievable by daily intake of 500 mg EPA + DHA supplement or by consuming fatty fish at least twice a week. There are many reports suggesting that the composition of FAs in lipid membranes seems to be influenced by the diet and/or supplements [8,9].

Shark liver oil (SLO) supplementation has been widely used recently in the prevention and treatment of diseases in humans. There are numerous studies that show how fish oil treatment may influence different types of dyslipidemias [9]. SLO is purified from sharks like Centrophorus squamosus, Cetorhinus maximus, Squalus acanthias, basking and dogfish sharks that live in cold, deep oceans. Certain compounds of SLO, such as alkylglycerols (AKGs), squalene and omega-3 PUFAs, are thought to be responsible for the low incidence of cancer in sharks themselves [10]. SLO has been traditionally used by Scandinavian fishermen for the treatment of a variety of ailments, including wounds, heart disease and infertility [10]. At present, SLO-based supplements are commercially available and consumed by different groups mostly due to their potential to enhance immunity. Research on SLO attributes its multiple health benefits to the high content of AKGs, squalene, lipid-soluble vitamins and PUFAs. AKGs are ether-linked glycerols containing such substances as batyl alcohol (AKG18/0), chimyl alcohol (AKG16/0) and selachyl alcohol (AKG18/1) [11,12]. AKGs are considered to be important immune-stimulating factors, able to increase the number of leucocytes, lymphocytes and platelets [10]. In vitro and in vivo studies suggest that AKG may offer anti-tumor potential via activating macrophage and exerting anti-angiogenesis effects [13,14,15]. Aside from activating macrophages, AKGs stimulate antibody production and enhance the function of Fc-receptors in the immune system [12,16]. No less significant role for health is demonstrated by the squalene. Some studies attribute squalene with anti-atherosclerotic effects [17,18]. Squalene is a precursor to cholesterol synthesis, accumulates in the liver and alters cholesterol and triglyceride synthesis [18,19]. It was shown that squalene may protect against chemically induced skin, colon and lung cancer due to its antioxidant action [20]. Furthermore, squalene is believed to reduce inflammation caused by anticancer drugs, making it a suitable add-on therapy during chemotherapeutic treatment [21]. Due to squalene’s immunostimulating activity, when used as an adjuvant, antibody levels significantly increased and inflammation slightly decreased [22]. Lastly, PUFAs, mostly omega-3 FAs that are also components of SLO, may influence the body’s immune function due to their anti-inflammatory effects [23]. As described earlier, the atheroprotective effect of omega-3 intake was associated with a reduction of VLDL synthesis and of small LDL particles [24]. An increased level of HDL-C fraction was also described in earlier studies where the omega-3 component of SLO was used as a supplement [25,26]. Additionally, it was pointed out in an animal study that omega-3 supplementation influences the structure and size of HDL-C that are in this way more effective in cholesterol reverse transportation [27]. Not surprisingly, the omega-3 PUFAs present in oils were described as factors that alter erythrocyte membrane FA concentration [28] and reduce the risk of heart disease [29,30].

Taking into account the possible influence of SLOon important aspects of health, we analyzed the impact of the short-term SLO supplementation on FAs metabolism in human erythrocytes and on some biochemical parameters of blood in the group of young healthy recipients. Our results showed that 6 weeks of SLO supplementation led to a significant increase in the content of both omega-3 and omega-6 PUFAs while reducing the content of saturated FAs in the erythrocyte membranes. From this, we observed significant alteration of the omega-3 index, as well as the membrane fluidity index. In addition, we found a noteworthy reduction in intracellular cholesterol level in PBMCs, as well as some decrease in C-reactive protein level in sera after SLO supplementation. In conclusion, our results showed that even short SLO supplementation can improve the erythrocyte FA composition, decrease inflammation and reduce cholesterol expression, which, all together, may have a favorable effect on human health.

## 2. Materials and Methods

### 2.1. Subjects and Supplementation Design

This study recruited 30 healthy, not smoking female volunteers of polish origin, at the age of 20–27 years (mean age 23.9 years). All volunteers provided written consent to participate in the SLO supplementation trial. The experiment was based on the intake of 1500 mg SLO per day for 6 weeks. All volunteers received nutritional education on SLO supplementation and were asked to maintain their usual diet during the period of the experiment. The commercially available supplement named Vitamarin, (Medana Pharma SA, Sieradz, Poland), was used in this experiment as a source of SLO. The study was conducted in accordance with the guidelines from the Declaration of Helsinki, and all procedures involving human participants were approved by the Bioethical Committee of Medical University of Lodz (RNN/22/16/KE).

### 2.2. Analysis of Serum Parameters

Blood samples, collected from volunteers at the beginning of the experiment and after 6 weeks of SLO supplementation, were used for FA determination (see below) and as the source of sera and peripheral blood mononuclear cells (PBMCs). The serum level of glucosetriacylglycerides (TAG), total cholesterol (TC), high-density lipoprotein-cholesterol (HDL-C) and low-density lipoprotein-cholesterol (LDL-C) were measured enzymatically with a colorimetric method using Roche Cobas C501 analyzer (Roche Diagnostics GmbH, Mannheim, Germany) according to the manufacturer’s protocol instructions. The appropriate commercial assay kits (Roche, Mannheim, Germany) were used for measure: glucose—Cat. No. 04404483 190; TAG—Cat. No. 20767107 322; HDL-C—Cat. No. 07528566 190; LDL-C—Cat. No. 07005717 190; TC—Cat. No. 03039773 190. C-reactive protein (CRP) level was assessed using the latex-enhanced immunoturbidimetric assay kit (Cat. No. 20764930 322, Roche, Mannheim, Germany) and read in Roche Cobas C501 analyzer. Simultaneously, in 12 randomly chosen sera, IFNα concentration was measured using the Human IFNα ELISA kit (Gen-probe Diaclone, SAS). In the case of the six participants that decided to continue SLO supplementation for additional 6 weeks, LDL-C concentration was determined a third time, after 12 weeks of SLO supplementation.

### 2.3. Analysis of Intracellular Cholesterol in PBMCs

Blood centrifugation on a density gradient (Biocoll 1077, Biochrom, Cambridge, UK) was used to isolate PBMCs. The relative intracellular cholesterol (IC) level in PBMCs was determined in all participants before and after 6 weeks of SLO supplementation. IC level was evaluated using the cholesterol assay kit, Cholesterol Chod-PAP (BIOLABO S.A., Maizy, France) per the manufacturer’s recommendations. These results were then normalized according to the protein concentration of each PBMC sample that was determined by Bradford assay (Bio-Rad, Hercules, CA, USA), as previously described [31].

### 2.4. Determination of FA Composition in Erythrocytes

Determination of erythrocyte membranes’ total FA composition was performed by a contract laboratory (Vitas Inc., Oslo, Norway) using gas chromatographic analysis (GC-FID). Blood samples were collected from all participants before and after 6 weeks of SLO supplementation, at the same time as the other samples. Internal standard (triheptadecanoin) was added and samples were methylated with 3N MeOH HCl. FAMEs were extracted with hexane, then samples were neutralized with 3N KOH in water. After mixing and centrifuging the hexane phase was injected into the GC-FID. Analysis was performed on a 7890A GC with a split/splitless injector, a 7683B automatic liquid sampler and flame ionization detection (Agilent Technologies, Palo Alto, CA, USA). Separations were performed on a TR-FAME (30 m × 0.25 mm i.d. × 0.25 µm film thickness) column from Thermo Fisher Scientific.

### 2.5. Statistical Analysis

Statistical analyses were performed with STATISTICA 8.0 PL software (StatSoft). The normality of variables was verified by the Shapiro–Wilk test. The results were analyzed by paired Student’s t-test or Wilcoxon matched-pairs signed rank test depending on the normality of the data distribution. Differences were considered statistically significant at *p*-values < 0.05. Data are presented as mean ± standard deviation (SD).

## 3. Results

### 3.1. Impact of SLO Supplementation on Biochemical Parameters of Sera

We investigated here the impact of short-term SLO consumption (1500 mg of SLO per day for 6 weeks) on some biochemical parameters in sera of our recipients. Neither BMI (21.18 vs. 21.16, *p* = 0.51) nor waist circumference (70.1 cm vs. 69.4 cm, *p* = 0.13) changed as a result of SLO supplementation. Concerning glucose concentration, it remained, as presented in Table 1, at similar levels before and after SLO consumption (88.33 vs. 88.29 mg/dL; *p* = 0.98).

Similarly, we did not find the significant differences (Table 1) in TAG concentration (69.9 vs. 73.4 mg/dL; *p* = 0.38) nor in the level of total cholesterol (162.37 vs. 158.81 mg/dL; *p* = 0.96) or in HDL-C fraction (68.14 vs. 68.27 mg/dL; *p* = 0.62). Concerning LDL fraction of cholesterol (Table 1) in sera, we could observe a significant increase of its concentration after 6 weeks of supplementation (from 76.6 mg/dL to 86.7; *p* = 0.003). In the case of the six participants that decided to continue SLO supplementation for an additional 6 weeks, we could evaluate LDL-C concentration after 12 weeks (Figure 1). In this group, the LDL-C level after 12 weeks did not differ significantly in comparison to the LDL-C level before supplementation (79.18 ± 14.61 mg/dL before SLO supplementation vs. 84.0 ± 15.44 mg/dL after 12 weeks of SLO supplementation; *p* = 0.506). However, we observed (Table 1) in all participants a small, but significant reduction of CRP in sera (1.78 mg/L vs. 1.51 mg/L after supplementation, *p* = 0.014). Conversely, the mean level of IFNα concentration, evaluated in the group of 12 participants, was higher after supplementation compared to the pre-supplementation state (8.1 pg/mL vs. 18.3 pg/mL, *p*= 0.003).

### 3.2. Impact of SLO Supplementation on Intracellular Cholesterol Level in PBMCs

Simultaneously, after 6 weeks of SLO consumption, the relative level of intracellular cholesterol (Figure 2), determined in PBMCs samples, was significantly decreased (0.95 ± 0.61 vs. 0.21 ± 0.21, *p* = 0.00003).

### 3.3. The Erythrocyte FAs Profile before and after SLO Supplementation

We analyzed the impact of short-term SLO consumption on FA composition in human erythrocyte membranes (both plasma and intracellular membranes). As presented in Table 2, six weeks of SLO supplementation had significantly influenced the FA composition. The relative contents of palmitic acid, as well as stearic acid, were significantly reduced after supplementation in comparison to the state before SLO administration (26.9 vs. 25.9 and 14.38 vs. 13.23, respectively).

By way of contrast, after the same period of time, we could observe a significant increase in the content of PUFAs from both omega-3 and omega-6 groups. In the first group (Figure 3A) the highest increase of relative content, from 2.78% to 3.51%, was found for the DHA, for DPA from 1.09% to 1.28% and for ALA from 0.41% to 0.46%. In the omega-6 group of PUFA (Figure 3B) we observed a significant increase in AA and DHGLA (≈12% and 7%, respectively) content. The content of the monounsaturated OA in erythrocytes did not change due to the SLO supplementation (21.95 vs. 21.77; *p* = 0.68).

The omega-3 index was estimated according to Harris et al. [6], by the sum of the relative percentage of EPA and DHA in erythrocytes. As presented in Figure 4A, six weeks of SLO supplementation caused a significant increase in the omega-3 index from 3.6% (before) to 4.2% (*p* = 0.0015). We observed also the impact of SLO supplementation on the membrane fluidity index, measured by the ratio between saturated and unsaturated FAs. A significantly important decrease of this index from 13.1 to 9.9 (*p* = 0.0014) is presented in Figure 4B.

## 4. Discussion

In the present study, we compared the blood lipid status before and after 6 weeks of SLO supplementation. To the best of our knowledge, no data concerning the blood lipid alteration after a short SLO supplementation in young, healthy women were reported. SLO beneficial/side effects were investigated mostly in animal models, as well as in human disease models, often using higher doses of SLO and longer-term oil consumption [12,32,33,34,35,36]. In this study, despite the short period of supplementation, we can observe some alteration of lipid metabolism. While the changes in serum lipid profile were not so pronounced, the composition of FAs in erythrocyte was significantly altered after supplementation. First of all, this short SLO supplementation changed the ratio between saturated and unsaturated FAs. A similar increase of unsaturated FA content in erythrocytes was observed after extra virgin olive and palm oil supplementation in another study [37]. Due to the significant increase of PUFA and some decrease of saturated FA content, the membrane fluidity index was improved. More than 20 years ago, it was reported that the reduced content of PUFA and the increased concentration of saturated FAs caused erythrocyte membrane stiffness [38]. In turn, the study of Hamadate et al. [35] presented the vascular effects of SLO supplementation on arterial stiffness. Eight weeks of SLO consumption improved the central arterial elasticity and peripheral microvascular function. No less important effect of SLO supplementation, in our study, was a marked elevation in the content of omega-3 (ALA, DPA and DHA) FAs in membranes, higher than the increase of omega-6 FAs (DHGLA, AA). Due to the increase of omega-3 PUFA, mostly DHA, an elevation of the omega-3 index was observed after 6-week SLO supplementation. This increased value of the omega-3 index is extremely desirable, but even this slightly elevated value is far from the one that is considered optimal for the proper functioning of the cardiovascular system [6]. This low value could be related to lower consumption of fish in Poland [7] in comparison to the mean intake in Europe or even in the world. Concerning serum lipid profile, no significant changes were observed in the level of TAGs, TC or HDL-C and LDL-C fraction after 6 weeks of SLO supplementation, which remains in accordance with the study of Hamadate [35]. However, other studies concerning this issue have provided some contradictory results. It was suggested that SLO or AKGs alone may increase the level of serum cholesterol and triglyceride [10,16,33]. Interestingly, in the study of Lewkowicz et al. [33], an increase of total cholesterol level and a decrease of HDL-C fraction were noted after 4-week SLO supplementation. However, the lipid metabolism normalized, after the end of the study, in all individuals. It is worth noting here that squalene, a major component of SLO, is a well-known precursor of cholesterol synthesis and can be responsible for a temporary increase of serum levels of cholesterol. However, results of previous studies [39,40] have reported that exogenous squalene rather inhibits 3-hydroxy-3-methylglutaryl coenzyme A (HMG-CoA) reductase, a key enzyme for cholesterol synthesis and orally administrated SLO, or a dietary source of squalene did not increase serum level of cholesterol [35]. Moreover, we observed (unfortunately, on a small number of samples) that the increase in serum LDL-C levels after 6 weeks of SLO supplementation decreases significantly after the next 6-week period of SLO administration. Similarly, it has been shown that a longer (3-month long) palm oil supplementation, decreases the LDL fraction and total cholesterol content in human plasma [37]. On the other hand, if we take into account the influence of omega-3 on lipid metabolism, we could also expect, as an effect, the decrease of triglycerides and triglyceride-rich lipoproteins synthesis by hepatocytes [41]. Similarly, an 11-week study of rat supplementation with squalene showed an increase in HDL cholesterol levels [17]. It is possible that in case of our study, short period of SLO administration and the low doses of SLO used for supplementation, can explain the lack of alteration of these important parameters.

Moreover, SLO administration markedly decreased the intracellular cholesterol expression measured in PBMCs of recipients. To our knowledge, there have not been any previous studies that explored the status of intracellular cholesterol under SLO supplementation. However, given the reports of lowering total cholesterol and LDL-C levels after prolonged SLO supplementation, we can assume that this significant alteration of cholesterol expression within the PBMCs may reflect the process taking place in the body after SLO supplementation. It was demonstrated that PBMCs profiling reflects gene expression levels of various tissues [42], which confirms previous observations concerning the close similarity in the expression of cholesterol between PBMCs and the liver [43], the main organ responsible for the regulation of cholesterol metabolism. As was mentioned above, our work showed that the decrease of serum LDL-C level is followed by its significant increase and it could be associated with the observed earlier decreasing of intracellular cholesterol expression. We can assume that the increase of serum IFN-alpha concentration after SLO supplementation in our study can be also associated with the alteration of lipid metabolism but the detailed molecular mechanism of such impact requires further research. It was earlier demonstrated that IFNs participate in different steps of the mevalonate pathway and IFN signals are able to reprogram the cellular cholesterol metabolism to support the function of immune cells [44]. On the other hand, it was shown that IFN-α treatment decreases lipid levels in rat hepatocytes and affects the expression of the HMG-CoA reductase gene and protein [45]. In this context and in view of our results, it seems also reasonable to undertake studies on the response of other inflammatory cytokines such as TNF-α, IL-6 or IL-1β.

SLO supplementation in our study showed a positive effect on the reduced expression of CRP, the well-characterized marker of the inflammatory processes. The decrease of CRP concentration was observed after 6 weeks of SLO supplementation but our study did not point out which component of SLO contributed to this positive response. Similar observations can be found in the study of Palmieri et al. [12] in which a group of old age surgical patients were treated with alkylglycerols before surgery and compared with a similar but untreated group of patients. A significant decrease of CRP in the treated patients was there one of the other benefits connected with AKG administration. It is worth noting that clinical studies showed that an omega-3-rich diet also provides anti-inflammatory effects [46]. It was demonstrated that marine-derived omega-3 PUFA supplementation had a significant lowering effect on CRP [47]. In addition, it was recently demonstrated [48] that pro-inflamative macrophage polarization in human adipose tissue was significantly reduced by the increase of alpha-linoleate and EPA content of membrane phospholipids. It is well-known that PUFAs are metabolized into several mediators of different immune and inflammatory activities. The omega-3 FAs serve as the precursors for the synthesis of anti-inflammatory lipid mediators [49]. Thus, as observed in our study, the increased content of alpha-linolenic acid and ALA-derived metabolites (DPA, DHA) could have an anti-inflammatory effect. It is also important to consider the individual differences of recipients in response to different types of supplementation. The study of Rudkowska et al. [50] is an interesting illustration of the metabolomics and transcriptomic differences in response to the omega-3 PUFA supplementation. We appreciate the fact that our study gave a few results that indicate what type of metabolic changes can be expected after a short SLO supplementation.

The main limitation of this study is the small number of participants, which consequently influenced the fact that we did not apply inclusion and exclusion criteria and also did not standardize the diet of the participants. Therefore, a large-scale study is required to better clarify the relationship between SLO supplementation and FA alterations.

## 5. Conclusions

Changes in the lipid content of erythrocytes are one effect of the described SLO administration. Such supplementation may be of special importance in countries such as Poland where the average consumption of either fish or fish oil is far below the mean world value. This study does not answer which component of SLO contributes, in particular, to this positive alteration. Nevertheless, on the basis of the changes of the composition of FAs in erythrocytes, as well as the decrease of intracellular cholesterol expression in PBMCs and CRP in sera, it can be suggested that even short-term SLO supplementation is beneficial for the health improvement of the recipient.

## Figures and Tables

**Figure 1 nutrients-13-03329-f001:**
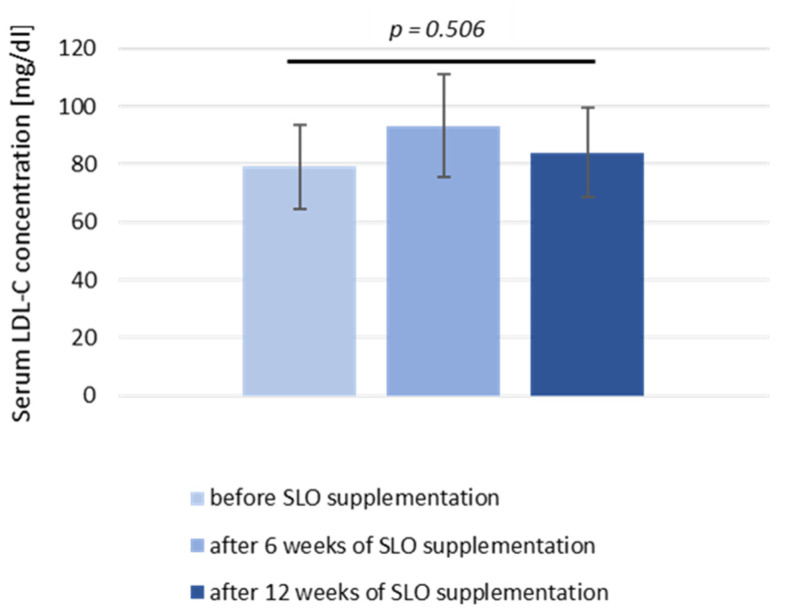
Serum LDL-C level in study participants before and after 6 and 12 weeks of SLO supplementation. Data are represented as mean ± SD in *n* = 6 subjects. Statistical analysis was performed by the Wilcoxon matched-pairs signed rank test.

**Figure 2 nutrients-13-03329-f002:**
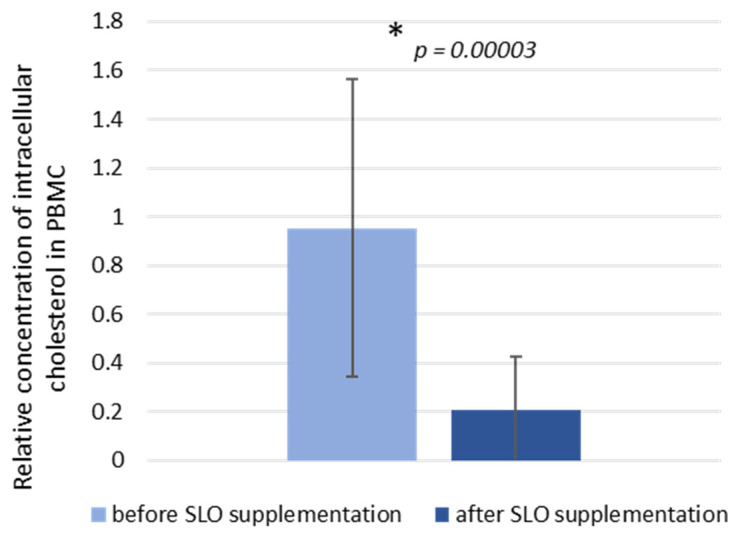
Relative concentration of PBMC intracellular cholesterol (the ratio of intracellular cholesterol to total cellular protein content) in study participants before and after 6 weeks of SLO supplementation. Data are represented as mean ± SD in *n* = 27 subjects. Statistical analysis was performed by the Wilcoxon matched-pairs signed rank test. * *p* < 0.05 was considered to be statistically significant.

**Figure 3 nutrients-13-03329-f003:**
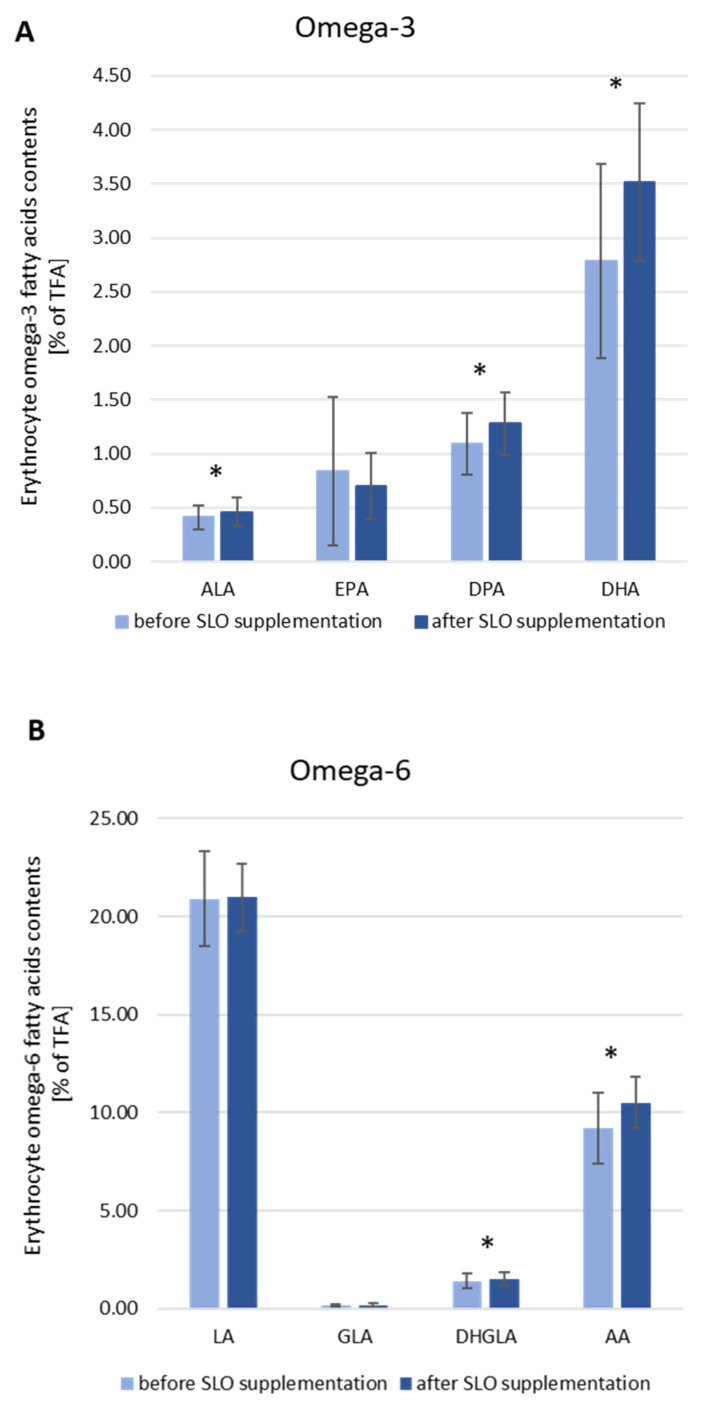
The erythrocyte content of omega-3 FAs (**A**) and omega-6 FAs (**B**) in study participants before and after 6 weeks of SLO supplementation. Data are represented as media ± SD in *n* = 27 subjects. Statistical analysis was performed by paired Student’s t-test (LA, DHGLA, DHA) or Wilcoxon matched-pairs signed rank test (EPA, DPA, ALA, GLA, AA). * *p* < 0.05 was considered to be statistically significant.

**Figure 4 nutrients-13-03329-f004:**
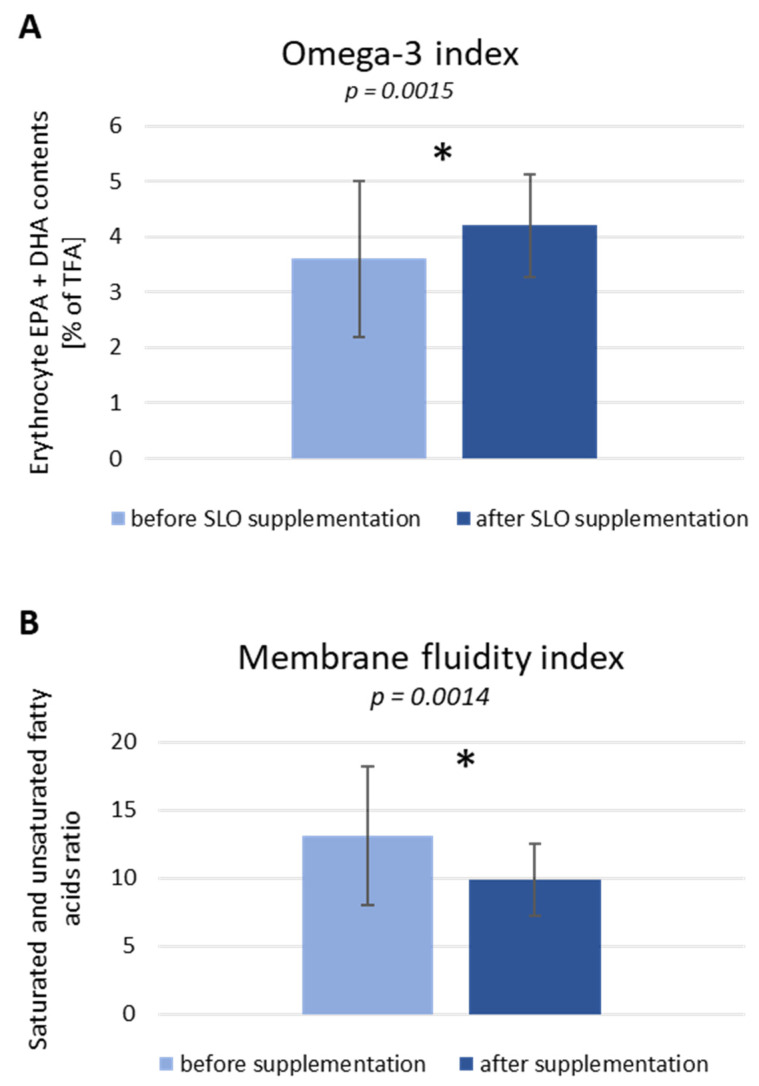
Omega-3 index (**A**) and membrane fluidity index (**B**) in study participants before and after 6 weeks of SLO supplementation. Data are represented as media ± SD in 25 subjects in the case of the omega-3 index and in 27 subjects in the case of membrane fluidity index. Statistical analysis was performed by Wilcoxon matched-pairs signed rank test (omega-3 index) and paired Student’s t-test (membrane fluidity index). (* *p* < 0.05) was considered to be statistically significant.

**Table 1 nutrients-13-03329-t001:** Measurements of serum biochemical parameters in study participants before and after 6 weeks of SLO supplementation. (*n*) value represents the number of subjects in which the analysis was performed.

Clinical Parameters	Before SLO Supplementation	After SLO Supplementation	*p* Value
Glucose; mg/dL (*n* = 24)	88.33 ± 8.79	88.29 ± 8.41	0.981
CRP; mg/L (*n* = 27)	1.78 ± 3.03	1.51 ± 2.67	0.014 (*)
TAG; mg/dL (*n* = 27)	69.96 ± 48.01	73.41 ± 32.02	0.388
TC; mg/dL (*n* = 27)	162.37 ± 28.98	158.81 ± 20.23	0.959
HDL-C; mg/dL (*n* = 27)	68.15 ± 16.11	68.27 ± 11.25	0.629
LDL-C; mg/dL (*n* = 26)	76.61 ± 23.05	86.65 ± 23.89	0.003 (*)
IFN-α; pg/mL (*n* = 18)	8.1 ± 9.6	18.3 ± 16.4	0.003 (*)

Data were expressed as mean ± SD; *p*-value indicates the level of statistical significance of paired Student’s *t*-test (Glucose, LDL-C) or Wilcoxon matched-pairs signed rank test (CRP, TAG, TC, HDL-C, IFN-α). * *p* < 0.05 was considered to be statistically significant.

**Table 2 nutrients-13-03329-t002:** Erythrocyte FA profile in study participants before and after 6 weeks of SLO supplementation.

Fatty Acid [% of TFA)	FA before SLO Supplementation (*n* = 27)	FA after SLO Supplementation (*n* = 27)	*p* Value
palmitic acid (PA)	26.92 ± 2.40	25.92 ± 2.09	0.020 (*)
stearic acid (SA)	14.38 ± 1.45	13.23 ± 0.92	0.000006 (*)
oleic acid (OA)	21.95 ± 2.36	21.77 ± 1.96	0.676
linoleic acid (LA)	20.90 ± 2.41	20.98 ± 1.71	0.828
dihomo-gamma-linolenic acid (DHGLA)	1.41 ± 0.39	1.50 ± 0.33	0.047 (*)
docosahexaenoic acid (DHA)	2.78 ± 0.90	3.51 ± 0.73	0.000017 (*)
eicosapentaenoic acid (EPA)	0.84 ± 0.69	0.70 ± 0.31	0.324
docosapentaenoic acid (DPA)	1.09 ± 0.29	1.28 ± 0.29	0.008 (*)
alpha-linolenic acid (ALA)	0.41 ± 0.11	0.46 ± 0.13	0.035 (*)
gamma-linolenic acid (GLA)	0.14 ± 0.07	0.17 ± 0.09	0.074
arachidonic acid (AA)	9.20 ± 1.80	10.51 ± 1.30	0.004 (*)

TFA—total fatty acids. Data were expressed as mean ± SD; *p*-value indicates the level of statistical significance of paired Student’s t-test (PA, SA, OA, LA, DHGLA, DHA) or Wilcoxon matched-pairs signed rank test (EPA, DPA, ALA, GLA, AA). * *p* < 0.05 was considered to be statistically significant.

## Data Availability

Data supporting reported results can be found at the Department of Medical Biochemistry, ul. Mazowiecka 6/8, 92-215 Lodz, Poland.

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
