# Peer review of "The Impact of Short-Term Shark Liver Oil Supplementation on the Fatty Acid Composition of Erythrocyte Membranes"

_nutrients, 2021, doi:10.3390/nu13103329_

Round 1

Reviewer 1 Report

The manuscript titled "The impact of the short shark liver oil supplementation on fatty acids composition of erythrocyte membrane" is well written and scientifically sound. The authors have cleared explained the need and significance of the study.

Following is my point by point comments:

  1. On page 171, the authors have mentioned that LDL-C level significantly decreased from 93.17 to 86.7 mg/dl but this is misleading as there is no significant difference with the control (before SLO supplementation). So, I would request the authors to change the bar graph to show the p-value in comparison to control only (before SLO supplementation)
  2. Interestingly, the authors also observed an increased level of IFN-a. Have they checked any other inflammatory cytokines such as TNF-a, IL-6, IL-1b and so on. It would be interesting to see their levels under such conditions.
  3. In the Discussion section, the authors speculate that increase in IFN-a is due to its association with lipid metabolism. However, it is not clear that how a change in lipid metabolism or SLO supplementation would affect IFN-a or other cytokines secretion. Please elaborate on this possible mechanism, as the SLO is upstream of cytokine secretion.
  4. Lastly, the data points are represented as a comma (,) instead of decimels(.). Please change that wherever needed.

Reviewer 2 Report

Dear Authors,

this paper is very interesting and can convey a positive message regarding the need for short shark liver oil supplementation. In my opinion, the methods of carrying out chemical analyzes are not well described. For example: determination of fatty acids composition in erythrocytes - "total fatty acids composition was performed by a contract laboratory (Vitas Inc, Oslo, Norway) using gas chromatographic analysis (GC-FID)".
I suggest that you specify the method of determination used (according to the author / standard?), The name of the chromatograph (producer).
